# Seroprevalence of Anti-SARS-CoV-2 Antibodies in a Random Sample of Inhabitants of the Katowice Region, Poland

**DOI:** 10.3390/ijerph18063188

**Published:** 2021-03-19

**Authors:** Jan E. Zejda, Grzegorz M. Brożek, Małgorzata Kowalska, Kamil Barański, Angelina Kaleta-Pilarska, Artur Nowakowski, Yuchen Xia, Paweł Buszman

**Affiliations:** 1Department of Epidemiology, School of Medicine in Katowice, Medical University of Silesia in Katowice, 40-752 Katowice, Poland; gbrozek@sum.edu.pl (G.M.B.); mkowalska@sum.edu.pl (M.K.); kbaranski@sum.edu.pl (K.B.); akaleta@sum.edu.pl (A.K.-P.); episars@sum.edu.pl (A.N.); pbuszman@ka.onet.pl (P.B.); 2State Key Laboratory of Virology, School of Basic Medical Sciences, Wuhan University, Wuhan 430072, China; yuchenxia@whu.edu.cn

**Keywords:** SARS-CoV-2, IgG/IgM seroprevalence, representative sample, general population, risk factors

## Abstract

Lack of knowledge around seroprevalence levels of COVID-19 in Poland was the reason for the implementation of a seroepidemiological study in the Katowice Region (2,100,000 inhabitants). In October–November 2020, a questionnaire examination and measurement of anti-SARS-CoV-2 IgG and IgM antibodies were performed in a random sample of the general population (*n* = 1167). The objectives of the study were to estimate the prevalence of IgG and IgM antibodies and to assess their host-related correlates. The prevalence of IgG seropositivity was 11.4% (95% CI: 9.5–13.2%) and IgM seropositivity was 4.6% (95% CI: 3.5–5.8%). Diagnosis of COVID-19 was found in 4.8% of subjects. A positive IgG test was statistically significantly associated with age (inverse relationship), a person’s contact with a COVID-19 patient, quarantine, and two symptoms in the past: fever and loss of smell/taste. Positive IgG tests were less prevalent in subjects who had diagnoses of arterial hypertension, diabetes, or rheumatologic disorders. IgM test positivity was associated with quarantine and loss of smell/taste only with no effect of chronic diseases found. In Poland, in the period October–November 2020, the prevalence of SARS-CoV-2 infection was larger than earlier estimates obtained in other European countries, probably reflecting the measurements obtained during the “second wave” of the epidemic.

## 1. Introduction

In Poland (pop. 37,672,367), the first confirmed case of COVID-19 was reported at the beginning of March 2020. Since then, up to 29 January 2021, the cumulative number of SARS-CoV-2 infections has reached 1,502,810 cases, including 36,780 deaths. At the beginning of the epidemic, the spread of infection was slow which could result from the population’s response to public health recommendations and regulations. The lock-down was imposed in March–April 2020, followed by gradual removal of restrictions in May–August. Summer appeared to change peoples’ attitude to public health regulations, and in September, children went back to school. Return of partial lock-down regulations took place in October and have been in effect since then. This flow of events parallels the changing epidemiology of COVID-19 in Poland. In the period between March–September, the daily number of reported SARS-CoV-2 infections increased from approximately 300 at the end of June to 500–600 in mid-September, and to the daily maximum of 27,875 in November. From the start of the pandemic until September, COVID-19 mortality usually did not exceed 30 deaths/day, and then sharply increased reaching 500–600 per day in November and December [1].

The evidence regarding COVID-19 incidence in Poland stems from nonsystematic observations and relies on the reports on newly detected infections as determined by RT-PCR testing of nasopharyngeal swabs, performed in selected hot spots and on medical indications. Above all, until November 2020 no serological testing was used to monitor the epidemic on a national or regional level and no epidemiological studies were implemented to assess the magnitude of the problem. To provide a serological assessment of the COVID-19 epidemic in Poland we designed a seroepidemiological study in the Katowice Region (Southern Poland). The primary objective of the study was to assess the prevalence of SARS-CoV-2 antibodies and to estimate the number of inhabitants infected with SARS-CoV-2 at the end of 2020. The secondary objective was to assess the host-related correlates of the presence of SARS-CoV-2 antibodies in the study population.

## 2. Materials and Methods

The study was performed as a cross-sectional seroepidemiological survey of a random sample stratified by age and sex. It was located in the Katowice Region which is a densely populated urban area (2,100,000 inhabitants), located in the center of the Katowice Voivodeship (administrative district). Within the region, three towns were chosen: Katowice (301,000 inhabitants), Gliwice (184,000 inhabitants), and Sosnowiec (209,000 inhabitants). The sample size was estimated assuming the expected 10% frequency of positive IgG tests, a 3% margin of error, and a 95% confidence level. Given the assumptions, the minimum sample size for each town was 384 subjects, resulting in a total minimum sample size required of 1152 subjects. Assuming limited participation, we decided to select an age-stratified sample of 2000 subjects in each town, using the central statistical database in Poland. The invitation letter was sent out in August–September and in October all selected persons received a second invitation. Of 6000 invited persons, only 1167 persons responded to the invitation (participation rate: 19.5%), 394 in Katowice, 392 in Gliwice, and 381 in Sosnowiec. Written informed consent was obtained from all participants, including parents of participants aged <18 years. The age distribution of the study group did not differ statistically (*p* = 0.8) from the age distribution of the general population of the Katowice Voivodeship (Figure 1).

In October–November 2020, all participants underwent questionnaire and laboratory examinations in local branch laboratories of consortium “Diagnostyka” located in Katowice (four locations per town, convenient working hours). The questionnaire included demographic questions as well as questions on data pertinent to COVID-19 (diagnosis, RT-PCR tests, quarantine, contacts), questions on symptoms suggestive of a viral infection in the period March–October/November 2020), and questions on a history of chronic diseases. Questionnaires were filled in by team members at the local laboratories.

At each laboratory, the certified nurses collected blood samples for the measurement of IgG and IgM antibodies. All measurements were performed in the central laboratory “Diagnostyka” using a semi-quantitative commercial ELISA test kit (EuroImmun Polska Sp z o.o, Wrocław, Poland). IgG and IgM antibodies were measured against S1 protein (IgG) and modified nucleocapsid protein (IgM) of SARS-CoV-2 in serum and the results were expressed as ratios (test/control extinction), according to the following scale: ratio < 0.8 = negative result, ratio 0.8–1.09 = questionable result, ratio > 1.09 = positive result. The product data sheet of the SARS-CoV-2 ELISA used in our study shows 94.4% sensitivity (10 days after symptom onset) and 99.6% specificity of the IgG test and 88.2% sensitivity (within 10 days after symptom onset) and 98.6% specificity of the IgM test [2]. Standard quality assurance procedures including calibration and control measurements were observed for each series of measurements not exceeding 100 samples.

Data analysis included simple and multivariate procedures. The Kolmogorov-Smirnov test was used to compare the age distributions of the sample and source populations. The distribution of quantitative variables was checked for normality using the Shapiro-Wilk test and was presented as the mean values and their standard deviations, and of categorical values as absolute and relative frequencies. Prevalence of positive IgG and IgM tests was presented by relative frequencies and their 95% confidence intervals (95% CI). Between-group differences in the distributions of quantitative variables were tested using the Wilcoxon test (data were not normally distributed), and for qualitative variables differences, the chi-square test was used. Associations between the qualitative variables were analysed using the chi-square test. Results of simple analyses were verified using multivariate logistic regression. The models included IgG test (+/−) or IgM test (+/−) as the dependent variable and history of COVID-19, sex, age, and symptoms as the independent variables. Parametrization of the models was performed using an automatic backward selection of explanatory variables. Their effects were assessed using odds ratios (OR) and corresponding 95% CIs. In the interpretation of the results of statistical analyses, *p* values below 0.05 were considered statistically significant. Statistical analyses were performed using procedures of the SAS 9.4 statistical package (SAS Institute, Cary, NC, USA).

The study was performed in line with the principles of the Declaration of Helsinki. Approval was granted by the Ethics Committee of the Medical University of Silesia in Katowice, Poland (file: PCN/0022/KB1/61/20; 14 July 2020).

## 3. Results

The study group included 1167 subjects, aged 0–94 years. The mean age was 46.2 ± 19.8 years and its distribution was not normal (*p* < 0.0001). The study group included 568 male subjects (48.6%) and 599 (51.4%) female subjects of similar age (46.7 ± 20.5 and 45.8 ± 19.2 years, respectively; *p* = 0.2). Among adult professionally active subjects (*n* = 635), 30 subjects (2.9%) had a medical profession. COVID-19 was diagnosed in the past in 56 subjects (4.8%), 142 (12.7%) had contact with a COVID-19 patient, 144 (12.3%) were put on quarantine, and 152 (13.0%) had an RT-PCR test in the past.

Table 1 shows the events relevant to COVID-19 infection and the occurrence of subjective symptoms experienced by the subjects since March 2020 by sex. Compared with male subjects, female subjects did not differ in COVID-19 related history but they had more frequent nonspecific symptoms suggestive of a viral infection.

In the study group, the reported diagnoses of chronic diseases included arterial hypertension (356 subjects, 30.5%), coronary artery disease (13 subjects, 1.1%), chronic obstructive pulmonary disease (37 subjects, 3.2%), asthma (83 subjects, 7.1%), diabetes (113 subjects, 9.7%), chronic allergic condition (116 subjects, 9.9%), oncologic disease (75 subjects, 6.4%), rheumatologic disorders (62 subjects, 5.3%), and autoimmunologic disorders (72 subjects, 6.1%).

Of 1167 subjects tested for SARS-CoV-2 IgG antibodies, 133 subjects were seropositive (11.4%; 95% CI: 9.5–13.2%), 1017 seronegative (87.2%), and 17 had a questionable test result (1.4%). The distribution of the results was similar (*p* = 0.3) in both sex groups: 10.2% in males, 12.5% in females. In terms of IgM antibodies, 54 subjects were seropositive (4.6%; 95%CI: 3.5–5.8%), 1095 seronegative (93.8%), and 18 had a questionable test result (1.6%). The distribution of the results was similar in males and females (4.1% and 5.2%, respectively; *p* = 0.2). The prevalence of IgG and IgM seropositivity did not differ between three locations (*p* = 0.2 and *p* = 0.7). Professionally active adults showed more IgG seropositivity than did unemployed adults (13.3% and 7.1%, respectively; *p* = 0.004) whereas both groups were similar in terms of IgM seropositivity (4.4% vs. 4.9%; *p* = 0.6).

Coexisting IgG and IgM seropositivity was found in 27 subjects (2.3%) and both negative results were found in 980 subjects (84.0%). Of 1167 tests, 466 (38.2%) were performed in October and 721 (61.8%) in November. Positive IgG tests obtained in October (5.6%) were less frequent (*p* < 0.0001) than those found in November (15.0%). In case of IgM the month-to-month difference was not statistically significant (*p* = 0.4): 3.5% and 5.3%, respectively.

Figure 2 shows the prevalence of positive IgG and IgM tests according to age category. The frequency of positive IgG tests depended on the age group (*p* = 0.005), whereas in the case of IgM tests, the effect of the age group was not statistically significant (*p* = 0.6). The largest prevalence of positive IgG tests was in older teenagers and young adults aged 15–35 years (18.1%; 95% CI: 13.3–23.6%). On the other hand, the largest prevalence of IgM positives was in the subjects aged 66+ years (6.0%; 95% CI: 2.9–9.2%).

Figure 3 shows the prevalence of positive IgG tests according to age group and sex. In males, the between-age difference was not statistically significant (*p* = 0.1), whereas, in females, the difference was statistically significant (*p* = 0.03). In males, no positive readings were found in the age category 76–90 years, and in females, all subjects aged 0–6 years were seronegative in terms of IgG. The IgG seropositivity reached a peak frequency of 40.0% in girls aged 15–18 years, followed by a frequency of 23.1% in women aged 19–25 years.

Analysis of the association between IgG test results and variables expressing COVID-19-related history, symptoms, and history of chronic diseases showed a more frequent occurrence of positive IgG tests in subjects who had contact with a COVID-19 patient (28.8% vs. 8.8%; *p* < 0.0001), history of quarantine (34.0% vs. 8.2%; *p* < 0.0001), and RT-PCR in the past (29.6% vs. 8.6%; *p* < 0.0001). The diagnosis of COVID-19 in the past and a positive result of the RT-PCR test did not differentiate the frequency of the IgG positive test in a statistically significant way. Results of simple analyses were verified by multivariate logistic regression. After adjustment for sex the explanatory variables of the positive IgG test were age (OR = 0.98; 95% CI: 0.72–0.99), contact with the COVID-19 patient (OR = 2.01; 95%CI: 1.22–3.33), and quarantine (OR = 4.00; 95% CI: 2.48–6.43).

Positive IgG tests were more frequent in subjects who had experienced the symptoms of fever (24.3% vs. 9.1%; *p* < 0.0001), chills (22.8% vs. 9.4%; *p* < 0.0001), fatigue (16.7% vs. 8.4%; *p* < 0.0001), cough (15.6% vs. 9.3%; *p* = 0.001), clogged nose (15.0% vs. 8.6%; *p* = 0.0006), dyspnea/trouble breathing (16.6% vs. 10.6%; *p* = 0.02), headache (14.6% vs. 9.8%; *p* = 0.01), nausea (18.4% vs. 10.9%; *p* = 0.04), and loss of smell/taste (51.9% vs. 7.3%; *p* < 0.0001). Results of multivariate logistic regression showed that after adjustment for sex, age, and history of COVID-19, the only explanatory variables of the positive IgG test were two symptoms: fever (OR = 2.0; 95% CI: 1.3–3.2) and loss of smell/taste (OR = 11.8; 95% CI: 7.5–18.6). Positive IgG tests were less prevalent in subjects who had such chronic diseases as arterial hypertension 7.9% vs. 12.9%; *p* = 0.01), diabetes (4.4% vs. 12.1%; *p* = 0.01), and rheumatologic disorders (1.6% vs. 11.95%; *p* = 0.01).

A positive IgM test was more prevalent in subjects who had contact with a COVID-19 patient (8.7% vs. 4.0%; *p* = 0.01) and who were put on quarantine (11.8% vs. 3.6%; *p* < 0.0001). In a multivariate analysis, quarantine appeared to be the only explanatory variable for a positive IgM test (OR = 3.65; 95% CI: 1.99–6.69). The only symptom associated with an IgM test was a loss of smell/taste, with more positive IgM tests in subjects who suffered from that symptom (11.3% vs. 3.9%; *p* = 0.0006). The effect of this symptom on IgM positivity was confirmed by the results of the multivariate analysis (OR = 3.1; 95% CI: 1.6–6.2). No statistically significant associations were found between chronic diseases and IgM seropositivity.

## 4. Discussion

The principal objective of our study was to assess the prevalence of SARS-CoV-2 antibodies and to estimate the number of inhabitants of the region infected with SARS-CoV-2 at the end of 2020. The spread of our measurements over two months hampers the estimation of a point prevalence of SARS-CoV-2 seropositivity in the study population. Under the assumption that the estimated prevalence is based on overall IgG antibody detection (11.4%; 95% CI: 9.5–13.2%), the estimated number of infected people in the region’s population of 2,100,000 is between 199,500 and 277,200 people. However, when the estimation is based on the most recent month (721 subjects tested in November: IgG+ in 14.9%; 95% CI: 12.3–17.5%), the likely number of infected persons in the Katowice Region is between 258,300 and 367,500 people of all ages. The obtained figures are likely to underestimate the true prevalence of SARS-CoV-2 infection because of a natural waning of IgG antibodies. The population size of the Katowice Region represents approximately 5% of the total population of Poland. Given this figure, it could be judged that in the country, the total number of people infected with SARS-CoV-2 ranged between 5,166,000 and 7,350,000 by the end of 2020.

A systematic review of studies describing the prevalence of anti-SARS-CoV-2 antibodies (IgG/IgM) and published up to 14 August 2020 showed a lower estimate than that provided by our study. In European countries, the pooled seroprevalence rates varied between 0.36% in Greece and 15.0% in Sweden [3]. However, the last estimate was provided by a study of 213 people. The second-largest estimate comes from Italy where the seroprevalence rate in 2323 screened people was 7.27%. The only Eastern European country included in the cited meta-analysis was Hungary with a 0.66% seropositivity rate. In Germany, the estimated seroprevalence rate was 2.23%. The region-wise seroprevalence was 3.17% in Western Europe, 4.41% in Southern Europe, and 5.27% in Northern Europe. Another systematic review and meta-analysis posted in November 2020 reviews the results of 281 serosurveys [4]. The majority of surveys included in the meta-analysis reported IgG tests (83%) and the review showed a median seroprevalence of 3.2% in the general population worldwide, larger in healthcare workers and caregivers (6.3%). A Belgian study, including a representative sample of the general population, conducted between March and July, showed a within-country range of seropositive cases between 2.9% and 6.9% [5]. Another population-based study in Geneva (Switzerland) showed that in the period 6 April–9 May 2020, the estimated weekly seroprevalence increased from 4.8% in the first week to 10.8% in the last week, the last figure being close to our estimate. As in our study, the Swiss findings were based on IgG assessment using a commercially available ELISA (Euroimmun) [6]. An increase in SARS-CoV-2 seroprevalence over two months (May–June, 2020) was also found in population-based studies in France [7]. In the region Ile-de-France, the prevalence of positive IgG reached 10% and was higher in young adults. In Europe, many seroepidemiological studies have focused on ‘hot-spots’ or ‘high-risk groups’. The published body of evidence regarding the general population provides only a few estimates with the apparent lack of data for Eastern Europe [3,4,8]. The apparent difference between our estimates of SARS-CoV-2 seroprevalence and estimates provided by general population studies published in the literature reflects the natural history of the COVID-19 pandemic. Our study was performed during the “second wave” of COVID-19 in Poland, whereas the published reports on serosurveys in general populations usually include results obtained in the first half of 2020. This fact may explain the larger seroprevalence observed in the Katowice Region compared to the estimates obtained earlier in other European countries.

Between-population variation in seropositivity of SARS-CoV-2 is likely to reflect differences in duration and phase of the epidemic, the natural waning of antibodies, socio-economic status, demographic and occupational factors, public health responses in terms of regulations and observance thereof affecting transmission dynamics [9,10]. In our study, the immunological tests were performed in October and November 2020. It is possible that a larger frequency of positive IgG tests in November as compared to October reflects the progress of the “second wave” of the COVID-19 epidemic in Poland. Such an interpretation is supported by other findings. Over a two month period there was an increase of subjects’ contact with COVID-19 patients from 7.6% to 15.9% (*p* < 0.0001), quarantine from 7.8% to 15.1% (*p* = 0.0002), and of diagnosed COVID-19 in the study group from 3.3% to 5.6% (*p* = 0.07).

Our major finding concerning seroprevalence of SARS-CoV-2 infection is based on the measurement involving IgG antibodies. Although IgG-mediated antibody positivity to SARS-CoV-2 is known to decline with time, using the IgG test to estimate the burden of SARS-CoV-2 infection is justified by better persistence of IgG antibodies in serum compared to other specific antibodies [11,12].

The secondary objective of our study was to assess the host-related correlates of the presence of SARS-CoV-2 antibodies. Our findings showed that the frequency of positive IgG results was inversely associated with age. This finding is in line with the conclusion of a recent systematic review that confirmed a larger seroprevalence in people aged 18–64 years compared to the age group 65 years or above [4]. A Swiss study also showed that young children and older people (≥65 years) had significantly lower seroprevalence than the other age groups [6]. In our study, the effect of age was also seen with the largest prevalence of IgG seropositivity in young subjects in contrast with the largest prevalence of IgM seropositivity in the elderly. The interpretation of this “mirror image” could take into account the difference in exposure circumstances. It might be speculated that young people experienced exposure to SARS-CoV-2 mostly during summertime and after return to school (i.e., 1–3 months before testing). Such an explanation favors IgG seropositivity, which is the marker of infectious exposure in the past. In contrast with young people, the elderly tended to observe a more restrictive lifestyle during the epidemic. However, some of them (6%) were not able to avoid SARS-CoV-2 infection during the “second wave”, as suggested by IgM seropositivity, the marker of a recent infection.

Although our study did not show a general effect of sex, and male and female subjects had a similar frequency of positive IgG tests, there was an apparent difference in age-related IgG positivity between the two sex groups. In comparison with male subjects who had a relatively homogenous distribution of IgG positivity over the age groups, the female subjects had the most prevalent positive IgG tests in the age range 7–25 years. The between-sex difference in the profiles of age-related distribution of IgG seropositivity is difficult to explain given our study protocol. However, in the age group 7–25 years and compared to males, the female subjects had more episodes of quarantine and two females but no males had a diagnosis of COVID-19. These differences were not statistically significant but might suggest a sex-related difference in past exposure to SARS-CoV-2 in response to differences in unidentified lifestyle factors. Nevertheless, the issue of sex-based biological mechanisms underlying the immune response to SARS-CoV-2 infection is understudied and deserves a comprehensive immunological assessment [13].

In our study, IgG positivity was also related to the person’s contact with COVID-19 patients and quarantine, which adds credibility to the results. Moreover, two symptoms experienced in the past were associated with positive IgG tests, namely fever and loss of smell/taste. The relatively strong statistical effect of the latter symptom corresponds with its diagnostic value in the natural history of COVID-19 [14].

An interesting finding was an inverse relationship of IgG positivity with the presence of hypertension, diabetes, and rheumatologic disorders. Such an association seems to support a view that people with chronic diseases tend to avoid every-day situations that may carry a risk of coronavirus exposure and better observe public health regulations. Such an explanation should also be discussed in terms of the finding that the increase in age (hence increasing occurrence of chronic diseases) was associated with lower IgG seropositivity. The interpretation should take into account the effect of age and, in particular, of chronic diseases on the immune status of older people. Our study design precluded insight into such mechanisms. With regard to the immune response and its modifiers, the body of evidence is imperfect and the issue deserves investigations into the immunogenicity of SARS-CoV-2, as recommended by the experts in the field [15,16]. Another interesting hypothesis is supported by more frequent (*p* < 0.0001) last year vaccinations against seasonal influenza in subjects aged above 65 years (21.3%) than in younger subjects (10.0%), in our study. The protective effect of influenza vaccination against COVID-19 was discussed in epidemiological reports, including the evidence obtained in the elderly [17,18,19]. Even if the association between the influenza vaccination and SARS-CoV-2 infection remains under investigation, such a protective effect could also result from a more restrictive lifestyle of vaccinated people, during the COVID-19 pandemic.

The results of ours and other European seroepidemiological studies involving the general population could be compared with pertinent evidence provided by population-based studies in China. In the former epicenter of COVID-19, at the end of the first wave of the pandemic in Wuhan, about 4% of people developed detectable IgG or IgM antibodies [20]. Ours and their results suggest that the number of infections was larger than the number of reported cases [21]. A comprehensive review of SARS-CoV-2 seroprevalence worldwide analyzed data obtained from a total of 86,000 people in China and showed a pooled seroprevalence of 1.63% in this country [3]. The estimates reported for Wuhan and other locations in China are much lower than the figures in our report. Several potential above-mentioned factors could explain the difference and the role of the current transmission of SARS-CoV-2 in the population or ethnicity should also be taken into account. Detection methods are also critical for the accuracy of the assessment of the population burden of COVID-19 [22]. Based on official Polish statistics, the cumulative prevalence of SARS-CoV-2 infection as assessed by RT-PCR testing in the study area over the same period was 2.8% [1]. Our study showed a cumulative seroprevalence of 11.4% thus suggesting that official statistics substantially underestimate the occurrence of SARS-CoV-2 infection. On the other hand, seroepidemiological studies may also underestimate the occurrence of SARS-CoV-2 infection [9,23]. For example, the underestimation may be linked with the performance characteristics of serological assays used in the study. Currently available serologic tests utilize different antigenic targets such as the S1 domain of the spike protein (S1), the recombinant nucleocapsid protein (NP), or the spike glycoprotein receptor-binding domain (RBD) [24]. In our study, we used a commercially available ELISA (Euroimmun) targeting the S1 domain of the spike protein of SARS-CoV-2 to detect IgG antibodies, and the modified nucleocapsid protein to detect IgM antibodies. The IgG assay used in our study has received FDA emergency use authorization (EUA) and shows good analytical performance, high sensitivity, and specificity [25]. Nevertheless, between-study comparisons should take into account the analytical aspects of the detection of SARS-CoV-2 seropositivity.

The principal conclusion of our report is based on the measurement of IgG antibodies, thus allowing the detection of infection at some point in the past. The estimation of the cumulative incidence of SARS-CoV-2 infection should not be based on the measurement of IgM antibodies which indicate recent infection and are not recommended to monitor the impact of coronavirus pandemic on the population [24]. Our findings seem to be in line with this view. The “second wave” of the COVID-19 epidemic in Poland started in September 2020 and our measurements showed that the frequency of positive IgG tests increased from 5.6% in October to 15.0% in November, whereas the increase of the frequency of positive IgM tests was not statistically significant (3.5% vs. 5.3%, respectively). It is not easy to explain the different performances of IgG and IgM tests over time, although both tests showed the same trend in terms of seropositivity. Two tests, IgG and IgM, have different detection profiles, they utilize different antigenic targets, and the IgM test has lower sensitivity and specificity compared with the IgG test. However, a more specific discussion on the reasons behind the statistically different seropositivity trends showed by IgG and IgM tests is not possible given our study protocol.

Our study has some limitations. The first limitation is the low participation rate. Active recruitment (two invitations/person) and wide promotion of the project in the media appeared to not be effective enough, probably because of peoples’ skepticism or fear of visits to laboratories during the epidemic. However, the final sample size met the estimated minimum sample size and was comparable with sample sizes of other population-based seroepidemiological studies. Another potential limitation is that we report seroprevalence estimates uncorrected for sensitivity and specificity of the analytical method. On the other hand, raw data can be easily used for between-project comparisons and transformations, if necessary.

The strengths of our study are its design including a random sampling method stratified by age and sex and its implementation before the start of the national vaccination program in Poland. Moreover, our study employed a questionnaire examination in addition to the measurement of antibodies and revealed some “questionnaire-antibodies” relationships that correspond with underlying plausible biological mechanisms.

## 5. Conclusions

To our knowledge, this is the first study to assess SARS-CoV-2 seropositivity in the general population in Poland. In this country, the prevalence of SARS-CoV-2 infection obtained in October-November 2020 was larger than earlier estimates obtained in other European countries, probably reflecting the ascending curve of the “second wave” of the epidemic. Such a conclusion seems to be supported by associations of anti-SARS-CoV-2 antibodies with a history of contacts with COVID-19 patients and symptoms suggestive of COVID-19. It is also of importance that this study’s estimates were obtained ahead of the national vaccination program initiated after the completion of our project. The study findings may serve as a baseline to monitor the evolution of the epidemic and to evaluate population response to vaccination.

## Figures and Tables

**Figure 1 ijerph-18-03188-f001:**
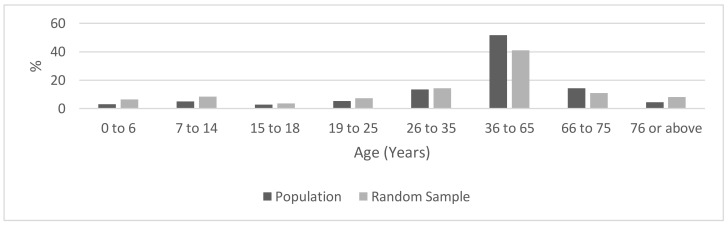
Age distributions of the random study sample and the source population.

**Figure 2 ijerph-18-03188-f002:**
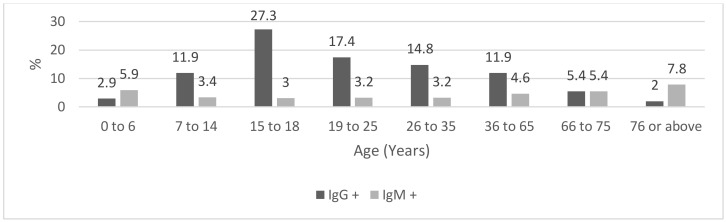
Prevalence (%) of positive IgG and IgM tests according to the age group.

**Figure 3 ijerph-18-03188-f003:**
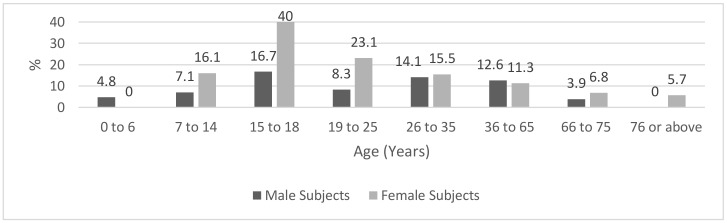
Prevalence (%) of positive IgG tests according to age group and sex.

**Table 1 ijerph-18-03188-t001:** History of COVID-19 infection, quarantine, molecular testing, and symptoms experienced in the period March-October in the study group, according to sex. The table shows the absolute and relative (% in the brackets) frequencies of the variables.

Variable ^a^	Males (*n* = 568)	Females (*n* = 599)	Total (*n* = 1167)	*p* Value ^b^
Diagnosis of COVID-19	29 (5.1%)	27 (4.5%)	56 (4.8%)	0.4
Contact with COVID-19 patient	70 (12.3%)	79 (13.1%)	149 (12.7%)	0.6
Quarantine	64 (11.3%)	80 (13.4%)	144 (12.3%)	0.2
RT-PCR test	74 (13.0%)	78 (13.1%)	152 (13.0%)	0.9
Fever > 38 C	83 (14.6%)	93 (15.5%)	176 (15.1%)	0.6
Chills	68 (11.9%)	103 (17.2%)	171 (14.6%)	0.01
Fatigue	181 (31.8%)	245 (40.9%)	426 (36.5%)	0.001
Sore throat	135 (23.7%)	206 (34.9%)	341 (29.2%)	0.0001
Cough	190 (33.4%)	194 (32.9%)	384 (32.9%)	0.6
Clogged nose	227 (39.9%)	278 (46.4%)	505 (43.3%)	0.02
Dyspnea	72 (12.6%)	85 (14.2%)	157 (13.4%)	0.4
Headache	143 (25.1%)	248 (41.4%)	391 (33.5%)	0.0001
Concjuctivitis	22 (3.8%)	27 (4.5%)	49 (4.2%)	0.7
Nausea	26 (4.5%)	50 (8.3%)	76 (6.5%)	0.009
Diarrhea	67 (11.8%)	72 (12.0%)	139 (11.9%)	0.9
Loss of smell/taste	36 (6.3%)	70 (11.7%)	106 (9.0%)	0.005

Legend: ^a^ all events/symptoms occurring prior to starting the study; ^b^ statistical significance level of the difference between males and females (the result of chi-square test).

## Data Availability

Data are available for research upon approval from Medical Research Agency, Poland. To obtain data researchers need to submit an analysis proposal to the corresponding author for evaluation and processing the request ith Medical Research Agency, Poland.

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
