# Peer review of "Seroprevalence of Anti-SARS-CoV-2 Antibodies in a Random Sample of Inhabitants of the Katowice Region, Poland"

_ijerph, 2021, doi:10.3390/ijerph18063188_

Round 1
Reviewer 1 Report
The manuscript's subject is important because its results allowed to identify the percentage of asymptomatic SARS-CoV-2 infected persons with the presence of IgG and IgM antibodies (about 5 times more than RT-PCR detected patients) and therefore it is a very useful study before vaccination campaign. The percentage of positives is in a good correlation with , for example, analogous study in St-Petersburg, Russia for May-June, 2020 which was recently published. The comparison with the data of Chinese situation is reasonable taking into account that the epidemy there was much shorter than in European countries. The limitation authors mentioned as low participation rate is really usual in such types of studies.
The shortage of the study is in usage of only one sort of test systems which is directed to the determination of S1-protein antibodies only. But it is well known that such test systems do not detect all the antibodies. Therefore it should be mentioned in DISCUSSION part of the manuscript.
Author Response
Dear Reviewer,
Thank you for your comment and recommendation. Following your advice we have added the following (below) text on the test systems, placed in the section ‘Discussion’.
Sincerely,
Jan E. Zejda
Response = New Text (Lines: 348-358)
For example the underestimation may be linked with performance characteristics of serological assays used in the study. Currently available serologic tests utilize different antigenic targets such as the S1 domain of the spike protein (S1), the recombinant nucleocapsid protein (NP), or the spike glycoprotein receptor-binding domain (RBD) [24]. In our study, we used a commercially available ELISA (Euroimmun) targeting the S1 domain of the spike protein of SARS-CoV-2 to detect IgG antibodies, and the modified nucleocapsid protein to detect IgM antibodies. The IgG assay used in our study has received FDA emergency use authorization (EUA) and shows good analytical performance, high sensitivity, and specificity [25]. Nevertheless, between-study comparisons should take into account the analytical aspects of the detection of SARS-CoV-2 seropositivity.
Reviewer 2 Report
The authors investigated the seroprevalence of anti-SRAS-CoV-2 antibodies in a region of Poland during the “second wave” of epidemic. This report is meaningful because it is important to assess and report on the seroprevalence of SARS-CoV-2 antibodies by country or region. However, this report seems to be inadequately considered based on the results.
Specific comments
- The authors described in Result section, “Positive IgGs obtained in October (5.6%) were less frequent (p <0.0001) than those found in November (15.0%).” Were there differences in the number of subjects diagnosed with COVID-19 or contacting with COVID-19 patients between subjects examined SARS-CoV-2 antibodies in October and those in November? Please indicate the comparisons in the text.
- The authors described in Result section, “The largest prevalence of positive IgG tests was in older teenagers and young adults aged 15 – 35 years (18.1%; 95%CI: 13.3 - 23.6%). On the other hand, the largest prevalence of IgM positives was in the subjects aged 66 + years (6.0%; 95%CI: 2.9 – 9.2%).” The authors should discuss about the difference of the age distribution between positive IgG and IgM antibodies.
- The authors should discuss that the age distribution of IgG-positive subjects differs between men and women.
- The authors considered about the reason of higher seroprevalence in this study in the 2nd and 3rd paragraphs of the Discussion section. However, it seems to be meaningless. Most of the cited studies were conducted before the “second wave” of epidemic in Europe. The results should be compared to those of studies investigated at the same period, and the authors should discuss based on the results.
- The authors considered that the reason the lower proportion of the seroprevalence in old people and people with chronic diseases tend to avoid every-day situations that might carry a risk of exposure to coronavirus. The reviewer recommended that the authors consider that, as other reason, old people and immunocompromised patients including those with diabetes and rheumatologic disorders have tended to have a low ability of the antibody production.
Author Response
Dear Reviewer,
We appreciate your comments and recommendations. Below please find our responses to each point. Please note, that with regard to the point #4 („comparisons”) we are able to offer only a minor addition to the text. Such a decision has some reasons. There is no doubt that the comparisons must be meaningful. The point is that at the time of data analysis and drafting the manuscript we made an extensive search for seroprevalence studies. As a result, we were were able to identify expert meta-analyses and other papers that showed the mid-2020 evidence at the best. More recent papers deal with specific populations (students, healthcare workers, blood donors) and are not good candidates for comparisons with estimates obtained in the general population. Our intention was not to highlight the difference between our results and published evidence but to reflect on the natural progress of the pandemic, resulting in the increase of seroprevalence with time. Following your recommendation, we made another up-to-date search and its results are not encouraging. One of the most recent reports (Lai, C.C.; Wang, J-H.; Hsueh, P-R. Population-based seroprevalence surveys of anti-SARS-CoV-2 antibody: An up-to-date review. Int. J. Infect. Dis. 2020, 101, 314-322; DOI: 10106/j.ijid.2020.10.011) is already included in our paper. Similarly, the important paper by Rostami, published in March (Rostami, A.; Sepidarkish, M.; Leeflang, M.M.G.; Riahi, S.M.; Shiadeh, M.N.; Esfandyari, S.; Mokdad, A.H.; Hotez, P.J.; Gasser, R.B. SARS-CoV-2 seroprevalence worldwide: a systematic review and meta-analysis, Clin Microbiol Infect 2020; DOI: 10.1016/j.cmi.2020.10.020) is also included (as a preprint) in our discussion. Other recent papers deal with blood donors in South Africa (preprint), describe a survey in Iran (17.1% in April-June), October survey in India (seroprevalence = 34%: Kar SS., Sarkar S, Murali S.et al.: Prevalence and time trend of SARS-CoV-2 Infection in Puducherry, India, August-October 2020 Emerg Inf Dis 2021; 27: 666-669 DOI 10.3201/eid2702.204480). In conclusion, we would be happy to refer to current or recent estimates of seropositivity in different general populations but the lack of pertinent data hampers our intention.
Following your comments and recommendations we added some pieces of text, shown below, and in the manuscript (the lines shown point by point)
Sincerely,
Jan E. Zejda
Specific comments
1.The authors described in Result section, “Positive IgGs obtained in October (5.6%) were less frequent (p <0.0001) than those found in November (15.0%).” Were there differences in the number of subjects diagnosed with COVID-19 or contacting with COVID-19 patients between subjects examined SARS-CoV-2 antibodies in October and those in November? Please indicate the comparisons in the text.
Response = New Text (Lines: 266-272 )
It is possible that a larger frequency of positive IgG tests in November as compared to October reflects the progress of the „second wave” of the COVID-19 epidemic in Poland. Such an interpretation is supported by other findings. Over a two month period there was an increase of subjects’ contact with COVID-19 patients from 7.6% to 15.9% (p<0.0001), quarantine from 7.8% to 15.1% (p=0.0002), and of diagnosed COVID-19 in the study group from 3.3% to 5.6% (p=0.07).
2.The authors described in Result section, “The largest prevalence of positive IgG tests was in older teenagers and young adults aged 15 – 35 years (18.1%; 95%CI: 13.3 - 23.6%). On the other hand, the largest prevalence of IgM positives was in the subjects aged 66 + years (6.0%; 95%CI: 2.9 – 9.2%).” The authors should discuss about the difference of the age distribution between positive IgG and IgM antibodies.
Response = New Text (Lines: 284-293)
In our study, the effect of age was also seen with the largest prevalence of IgG seropositivity in young subjects in contrast with the largest prevalence of IgM seropositivity in the elderly. The interpretation of this „mirror image” could take into account the difference in exposure circumstances. It might be speculated that young people experienced exposure to SARS-CoV-2 mostly during summertime and after return to school (i.e. 1-3 months before testing). Such an explanation favours IgG seropositivity, which is the marker of infectious exposure in the past. In contrast with young people, the elderly tended to observe a more restrictive lifestyle during the epidemic. However, some of them (6%) were not able to avoid SARS-CoV-2 infection during the „second wave”, as suggested by IgM seropositivity, the marker of a recent infection.
3. The authors should discuss that the age distribution of IgG-positive subjects differs between men and women.
Response = New Text (Lines: 294-306)
Although our study did not show a general effect of sex, and male and female subjects had a similar frequency of positive IgG tests, there was an apparent difference in age-related IgG positivity between the two sex groups. In comparison with male subjects who had a relatively homogenous distribution of IgG positivity over the age groups, the female subjects had the most prevalent positive IgG tests in the age range 7-25 years. The between-sex difference in the profiles of age-related distribution of IgG seropositivity is difficult to explain given our study protocol. However, in the age group 7-25 years and compared to males, the female subjects had more episodes of quarantine and two females but no males had a diagnosis of COVID-19. These differences were not statistically significant but might suggest a sex-related difference in past exposure to SARS-CoV-2 in response to differences in unidentified lifestyle factors. Nevertheless, the issue of sex-based biological mechanisms underlying the immune response to SARS-CoV-2 infection is understudied and deserves a comprehensive immunological assessment [13].
4.The authors considered about the reason of higher seroprevalence in this study in the 2nd and 3rd paragraphs of the Discussion section. However, it seems to be meaningless. Most of the cited studies were conducted before the “second wave” of epidemic in Europe. The results should be compared to those of studies investigated at the same period, and the authors should discuss based on the results.
Response = New Text (Lines: 254 - 259)
The apparent difference between our estimates of SARS-CoV-2 seroprevalence and estimates provided by general population studies published in the literature reflects the natural history of the COVID-19 pandemic. Our study was performed during the „second wave” of COVID-19 in Poland, whereas the published reports on serosurveys in general populations usually include results obtained in the first half of 2020.
5. The authors considered that the reason the lower proportion of the seroprevalence in old people and people with chronic diseases tend to avoid every-day situations that might carry a risk of exposure to coronavirus. The reviewer recommended that the authors consider that, as other reason, old people and immunocompromised patients including those with diabetes and rheumatologic disorders have tended to have a low ability of the antibody production.
Response = New Text (Lines: 318 – 330)
The interpretation should take into account the effect of age and, in particular, of chronic diseases on the immune status of older people. Our study design precluded insight into such mechanisms. With regard to the immune response and its modifiers, the body of evidence is imperfect and the issue deserves investigations into the immunogenicity of SARS-CoV-2, as recommended by the experts in the field [15, 16]. Another interesting hypothesis is supported by more frequent (p<0.0001) last year vaccinations against seasonal influenza in subjects aged above 65 years (21.3%) than in younger subjects (10.0%), in our study. The protective effect of influenza vaccination against COVID-19 was discussed in epidemiological reports, including the evidence obtained in the elderly [17-19]. Even if the association between the influenza vaccination and SARS-CoV-2 infection remains under investigation, such a protective effect could also result from a more restrictive lifestyle of vaccinated people, during the COVID-19 pandemic.
Reviewer 3 Report
This is a thorough, well-designed epidemiological assessment of seropositive status of the population of a region in Poland. The data are well presented and explained, and compared to existing findings in the region.
Minor comments:
Specificity of IgM test?
The lack of differences between sexes and in October vs November for IgM, where there are differences in IgG, brings the specificity of this test into question.
A number of grammatical errors, but not to the extent that the text was not understandable
Author Response
Dear Reviewer,
We appreciate very much your opinion regarding our manuscript. Thank you very much for the comments. In response to your advice we have added new texts, one in ‘Methods’, one in ‘Discussion’. Both additions are shown below and in the manuscript (‘Methods’, ‘Discussion’). I appreciate your gentle treatment of my ability to write in English. Following your remark I asked a native speaker for the refinement of text.
Sincerely,
Jan E. Zejda
Response 1 (‘Methods’) = New Text (Lines: 98-101)
The product data sheet of the SARS-CoV-2 ELISA used in our study shows 94.4% sensitivity (10 days after symptom onset) and 99.6% specificity of the IgG test and 88.2% sensitivity (within 10 days after symptom onset) and 98.6% specificity of the IgM test [2].
Response 2 (‘Discussion’) = New Text (Lines: 359-373)
The principal conclusion of our report is based on the measurement of IgG antibodies, thus allowing the detection of infection at some point in the past. The estimation of the cumulative incidence of SARS-CoV-2 infection should not be based on the measurement of IgM antibodies which indicate recent infection and are not recommended to monitor the impact of coronavirus pandemic on the population [24]. Our findings seem to be in line with this view. The „second wave” of the COVID-19 epidemic in Poland started in September 2020 and our measurements showed that the frequency of positive IgG tests increased from 5.6% in October to 15.0% in November, whereas the increase of the frequency of positive IgM tests was not statistically significant (3.5% vs 5.3%, respectively). It is not easy to explain the different performances of IgG and IgM tests over time, although both tests showed the same trend in terms of seropositivity. Two tests, IgG and IgM, have different detection profiles, they utilize different antigenic targets, and the IgM test has lower sensitivity and specificity compared with the IgG test. However, a more specific discussion on the reasons behind the statistically different seropositivity trends showed by IgG and IgM tests is not possible given our study protocol.
Round 2
Reviewer 2 Report
The authors revised the manuscript according to the reviewer's comments.